# Effects of Dispersant on the Petroleum Hydrocarbon Biodegradation and Microbial Communities in Seawater from the Baltic Sea and Norwegian Sea

**DOI:** 10.3390/microorganisms11040882

**Published:** 2023-03-29

**Authors:** Ossi Tonteri, Anna Reunamo, Aura Nousiainen, Laura Koskinen, Jari Nuutinen, Jaak Truu, Kirsten S. Jørgensen

**Affiliations:** 1Marine Research Centre, Finnish Environmental Institute, Latokartanonkaari 11, FI-00790 Helsinki, Finland; 2Laboratory Centre, Finnish Environmental Institute, Latokartanonkaari 11, FI-00790 Helsinki, Finland; 3Institute of Molecular and Cell Biology, Faculty of Science and Technology, University of Tartu, Riia 23, 51010 Tartu, Estonia

**Keywords:** oil biodegradation, dispersant, Baltic Sea, Norwegian Sea, oil spill, bacterial community

## Abstract

Dispersants have been used in several oil spill accidents, but little information is available on their effectiveness in Baltic Sea conditions with low salinity and cold seawater. This study investigated the effects of dispersant use on petroleum hydrocarbon biodegradation rates and bacterial community structures. Microcosm experiments were conducted at 5 °C for 12 days with North Sea crude oil and dispersant Finasol 51 with open sea Gulf of Bothnia and coastal Gulf of Finland and Norwegian Sea seawater. Petroleum hydrocarbon concentrations were analysed with GC-FID. Bacterial community structures were studied using 16S rDNA gene amplicon sequencing, and the abundance of genes involved in hydrocarbon degradation with quantitative PCR. The highest oil degradation gene abundances and oil removal were observed in microcosms with coastal seawater from the Gulf of Bothnia and Gulf of Finland, respectively, and the lowest in the seawater from the Norwegian Sea. Dispersant usage caused apparent effects on bacterial communities in all treatments; however, the dispersant’s effect on the biodegradation rate was unclear due to uncertainties with chemical analysis and variation in oil concentrations used in the experiments.

## 1. Introduction

In 2013, 315 million tonnes of liquid oil cargo were handled in Baltic ports, and more than 40% of this volume was transported across the Baltic Sea area between Finnish, Russian and Estonian ports located in the Gulf of Finland [1]. Maritime traffic in the Baltic Sea has been predicted to continue to increase until 2030 [2]. Increases in marine traffic and oil transportation have increased the risk of oil spills in the Baltic Sea region. Oil tanker accidents in the Gulf of Finland would have significant ecological consequences since the oil spill would likely reach vulnerable coastal areas.

The most common method for oil recovery during oil spills is mechanical recovery with different brushes and skimmers. However, mechanical recovery can only recover oil that forms a slick on the water’s surface. As an alternative option to recovery, chemical dispersants have been used in many parts of the world. Dispersants consist of surfactants that are dissolved in solvents, and they function by reducing the surface tension between oil and water and breaking the oil into small droplets, thus dispersing oil into the water phase. Dispersants may thus prevent the oil from stranding and reaching the shore. Dispersants are considered to increase bioavailability and accelerate natural biodegradation [3]. 

Although dispersants have not been used in the Baltic Sea (or the Gulf of Finland), it has been suggested that they could be used in limited quantities in cases where the oil spill would endanger a large number of overwintering birds and where other response measures would be limited by ice cover [4]. The use of dispersants is not forbidden in Finland, but their use is regulated by national legislation and requires permission from the competent response authority. Furthermore, the Helsinki Commission (HELCOM) does not currently recommend the use of dispersants in the Baltic Sea (HELCOM recommendation 22 February 2001).

Dispersants were used extensively during the Deepwater Horizon oil spill, where 1.83 million gallons of Corexit 9500a and Corexit 9527A were applied to the surface and injected into the wellhead to prevent crude oil from reaching the coastal ecosystems [5] Although dispersants have been used as an oil spill response method, there have been contradictory results on how they affect biodegradation and microbial communities. Some laboratory-scale studies have shown that oil biodegradation is enhanced with the application of dispersants [6,7,8], but there have also been studies that suggest that the dispersant chemicals could inhibit biodegradation [9,10], and some studies have reported that dispersants have no effects on biodegradation [11]. Dispersants have also been observed to either enhance or inhibit biodegradation potential, depending on the microbial strain used in the tests [12]. Rughöft et al. (2020) [13] observed that substrate limitation, similar to environmental conditions of the open sea, can affect the response and activities of oil-degrading microbes when exposed to Corexit dispersant.

There are a number of studies on crude oil biodegradation and the effects of oil pollution on microbial communities that have been conducted with Arctic seawater (e.g., [14,15,16]), but only a few have been undertaken with Baltic Sea seawater [17,18] However, the previous studies conducted with Baltic Sea seawater have assessed oil biodegradation without the addition of dispersants. In this study, microcosm experiments were carried out to study the effect of dispersant on the crude oil biodegradation rate and the bacterial community composition of Baltic Sea and Norwegian Sea microbial communities. Microcosm experiments were set up to examine degradation rates, and changes in microbial communities exposed to oil and dispersants were studied using 16S rDNA gene amplicon sequencing. In addition, the abundance of polyaromatic hydrocarbons (PAH) and alkane degradation genes was measured by quantitative PCR.

## 2. Materials and Methods

### 2.1. Seawater Sampling

Water samples were collected from the Gulf of Finland (Tvärminne Zoological Station, Finland, May 2017 and March 2018), Gulf of Bothnia (2017) and Norwegian Sea (Kvitvika, Ofotfjord) (Norway, June 2018) (Figure 1). Sampling locations and dates are presented in Table 1. All water samples were collected from ca. 1 m depth and stored at 5° C for a maximum of 5 days before starting the experiments.

### 2.2. Crude Oil and Dispersants

A microcosm experiment was set up to investigate crude oil biodegradation and changes in bacterial communities. The experiment was conducted using naphthenic North Sea (NNS) crude oil, which contains high amounts of low molecular weight hydrocarbons such as decalins and naphthalenes. To test effects of dispersants on the crude oil biodegradation and bacterial communities, Finasol OSR 51^®^ (Total Special Fluids, France) was used. According to the Safety Data Sheet for Finasol 51, it consists of docusate sodium (0.2–5 wt%) and hydrocarbons (60–70 wt%).

### 2.3. WAF and CE-WAF Preparation

In order to mimic relevant concentrations of oil and dispersed oil in a true situation in situ, we used water accommodated fractions of oil (Lee et al. 2013). In the experiments, crude oil was applied as water accommodated fractions (WAFs), and treatments with dispersant were applied as chemically enhanced water accommodated fractions (CE-WAFs). WAFs and CE-WAFs were prepared following [19]. Briefly, these were prepared by adding 5 g/L of crude oil to aspirator glass vessels containing unfiltered seawater and placed in shaker (WAF) or mixed with magnetic stirrer (CE-WAF) with low energy for 40 h at 10 °C in the dark. CE-WAFs were prepared by adding 1:10 ratio of dispersant into the crude oil before adding it to seawater. After settling for 2 h, water fractions were drained via a valve on the bottom of the flask, and ready WAFs and CE-WAFs were immediately used to start the microcosm experiment.

### 2.4. Microcosm Setup

Laboratory-scale microcosm experiments consisted of triplicate 1 L glass bottles that included either a WAF, a CE-WAF or unfiltered seawater as control. Because the dispersants cause the oil to disperse into the water phase, the concentrations of petroleum hydrocarbons are much higher in the CE-WAF than in the WAF. Microcosm experiments with Gulf of Finland seawater were conducted in three parts (Table 2), the first experiment with undiluted WAF and CE-WAF (abbreviated GoF high). The experiment with Gulf of Bothnia (GoB) seawater was performed using dilutions 1:1 (WAF) and 1:1000 (CE-WAF). The second Gulf of Finland experiment (GoF low) and the experiment with Norwegian Sea seawater (NS) were conducted with dilutions 1:1 (WAF) and 1:50 (CE-WAF).

The experiments were conducted at 5 °C in the dark, and bottles from all treatments were periodically sacrificed after 0 h, 24 h, 48 h and 12 or 13 days after the start of the experiment for biological and chemical analysis.

An additional control experiment was conducted separately to investigate the quantitative and qualitative properties of crude oil and which fractions are transferred to the water phase. This experiment was conducted as described above using seawater from the Gulf of Finland using dilutions 1:1 (WAF) and 1:50 (CE-WAF). The control experiment included samples containing only the dispersant (Finasol 51) and abiotic controls (WAF and CE-WAF treatments with HgCl_2_). Abiotic treatments were conducted in triplicate and all other treatments were conducted in duplicate.

### 2.5. Chemical Analysis

Analyses for petroleum hydrocarbons (fractions C_10_-C_40_) were conducted by the Finnish Environment Institute Laboratory Centre. Undiluted CE-WAF and WAF from the control experiment were analysed. From the microcosm experiments, three replicates from each timepoint and treatment were analysed. Abiotic controls were analysed only at the beginning and end of the control experiment. Quantification of petroleum hydrocarbons levels (fractions C_10_-C_21_, C_21_-C_40_ and C_10_-C_40_) was performed with gas chromatography with flame ionization detector (GC-FID; Shimadzu GC-2010 Plus AF) according to ISO 9377-2:2000 [20]

Samples were adjusted to pH of 2 with HCl and subsequently extracted with hexane as an extraction solvent which contained *n*-decane and *n*-tetracontane. Then, 30 mL of hexane was added to the water samples and those were extracted after 30 min with a magnetic stirrer. After extraction, the hexane layer was separated. If an emulsion was formed, it was removed by centrifuging the extract. The whole extract was purified with commercial Florisil cartridges (Chromabond Florisil, Macherey-Nagel) to remove polar and non-petrol compounds. Finally, the cartridge was rinsed with about 10 mL of hexane. The extract was concentrated slightly to 1 mL using a gentle flow of nitrogen, keeping the temperature at 40 °C. The final amount of extract was calculated by weighing. The aliquot of the final extract was transferred to a vial for gas chromatographic analysis.

PAH were analysed from undiluted and diluted control experiment CE-WAF and WAF samples. Furthermore, PAH were analysed in duplicate at 0 h and 12 days of the control experiment. PAH samples were analysed with GC-MS/MS after extraction with *n*-hexane (Fluka) and purified with a solid phase extraction (SPE) cartridge (Biotage, ISOLUTE EPH, 5 g) based on ISO 28540:2011 [21]. Analysis was conducted with a gas chromatography-triple quadrupole mass spectrometer GC-MS/MS (Thermo Scientific, Waltham, MA, USA). The GC-MS/MS analysis of PAH compounds was performed with a Trace 1310 GC Ultra gas chromatograph (Thermo Scientific) interfaced with a Thermo Scientific TSQ Quantum XLS ultra mass spectrometer (Thermo Scientific) in the electron impact (EI) mode. The GC separation was achieved using a selected PAH column (Agilent Technologies) (30 m × 0.25 mm × 0.15 µm). Altogether 21 PAH compounds were analysed [22].

### 2.6. DNA Extraction

DNA samples from the microcosm experiments were obtained by filtering the 1 L bottles using Steritop^®^ 0.22 µm Millipore Express^®^ PLUS membranes (Merck Millipore, Darmstadt, Germany). Filters were stored at −20 °C before extraction. DNA extraction was performed using a FastDNA^®^ SPIN Kit for Soil (MP Biomedicals, Santa Ana, CA, USA) according to the manufacturer’s protocol. Thawed filters were cut in two parts, and half of the filter was inserted into a bead tube for bacterial lysis. Supernatants from the two halves were combined into the same tube after the lysis step. Samples were homogenized with the FastPrep instrument (MP Biomedicals, Santa Ana, CA, USA) on setting 4.5 for 30 s.

### 2.7. Real-Time Quantitative PCR

Real-time PCR quantification was performed using a 7300 Real Time PCR System (Applied Biosystems, Thermo Fisher, Waltham, MA, USA). The abundance of alkane degraders (*alkB* gene) was quantified using *alkB* primer [23]. The abundance of PAH degraders (PAH-RHDα gene) was quantified using separate primers for gram-positive (PAH-RHDα GP) and gram-negative (PAH-RHDα GN) [24] (Table 3). The abundance of total bacteria (16S rDNA gene) was quantified with primers Eub338f [25] and Eub518r [26]. Reactions with PAH gram-positive and gram-negative primers were performed in 25 µL and 12.5 µL volumes with a Maxima Power SYBR^®^ Green PCR Master Mix (Thermo Scientific, Pittsburgh, PA, USA) and 5–10 µL of DNA template. Reactions with 16S and *alkB* primers were conducted in 25 µL reactions with a QuantiTect SYBR^®^ Green Master Mix (Qiagen, Hilden, Germany). Thermal cycling protocol for 16 S rDNA and *alkB* consisted of a denaturation step of 15 min at 95 °C followed by 45 cycles with 15 s at 94 °C, 30 s at 53 °C, 30 s at 72 °C and 27 s at 80 °C. Thermal cycling protocol for PAH-RHDα gram-positive consisted of a denaturation step of 10 min at 95 °C followed by 15 s at 95 °C, 30 s at 57 °C, 30 s at 72 °C and 27 s at 83 °C for the gram-negative, 10 min at 95 °C was followed by 15 s at 95 °C, 30 s at 54 °C, 30 s at 72 °C and 27 s at 86 °C.

Standards used in the analysis were *Escherichia coli* plasmids with PAH–RHDα inserts amplified and cloned from *Pseudomonas putida* G7 (GN) or *Mycobacterium vanbaalenii* DSM 7251 (GP) and from *Pseudomonas putida* (*alkB*). Calibration was completed using standards with gene copies of 10^1^ to 10^10^ µL^−1^.

### 2.8. 16 S rDNA Amplicon Preparation for Sequencing

The seawater bacterial community composition was assessed by Illumina^®^ MiSeq sequencing of combinatorial sequence-tagged PCR products using the universal barcoded [27] primers 515F and 926R targeting the 16S rRNA gene region [28]. Amplification of sample DNA was performed in triplicate in an Eppendorf Mastercycler PCR machine (Eppendorf, Hamburg, Germany). After pooling the replicate PCR products, the concentration was measured with a TapeStation 2200 using D1000 ScreenTapes^®^ (Agilent Technologies, Santa Clara, CA, USA) and samples were combined in equal proportions. The final library was then purified and concentrated using a NucleoSpin^®^ Extract II kit (Macherey-Nagel GmbH & Co. KG, Düren, Germany). The sequencing library was prepared by adaptor ligation and PCR using a TruSeq Nano DNA Library Prep Kit according to the producer protocol but excluding the fragmentation step [29]. The DNA library was sequenced on an Illumina^®^ MiSeq system (2 × 250 v2) (Illumina Inc., San Diego, CA, USA) at Microsynth AG (Balgach, Switzerland). Raw 16S rRNA sequences are available at the European Nucleotide Archive (ENA) under the study accession number PRJEB56358.

### 2.9. Sequence Data Analyses

Paired-end sequences assembly and demultiplexing of the sequences were conducted with PEAR version 0.9.11. Sequences were further analysed using the open-source platform Mothur v.1.48.0 [30] following the MiSeq SOP (https://www.mothur.org/wiki/MiSeq_SOP (accessed on 20 March 2023)). Sequences shorter than 360 bp were screened out. Sequences were aligned using the Silva reference database v. 138.1. Sequences with ≥97% similarity were assigned to Operational Taxonomic Units (OTUs) using the *vsearch* algorithm [31].

### 2.10. Statistical Analysis

Statistical analyses were carried out using Past 4.03 [32]. Differences in the rate of hydrocarbon degradation and qPCR copy numbers in the different experiments and treatments were determined using one-way ANOVA.

MicrobiomeAnalyst [33] was used to create principal coordinates analysis (PCoA) plots and a linear discriminant analysis effect size (Lefse) analysis to visualise the patterns of bacterial communities and their differences between treatments.

Microbial community differences between geographically different seawater and treatments were assessed using permutational multivariate analysis of variance (PERMANOVA). A heatmap was created using Clustvis [34].

## 3. Results

### 3.1. Oil Behavior in WAF and CE-WAF Preparations

Based on the chromatograms obtained from the additional experiment, not all fractions of crude oil were transferred entirely into the water phase, as WAF samples have very little n-alkanes present but included lighter (<15 C) compounds, most likely aromatic hydrocarbons and PAH compounds (Appendix A). Dispersant addition helped to dissolve the oil in the water phase as the pure crude oil samples and CE-WAF 0 h samples showed the similar chromatograms from the whole hydrocarbon range (Appendix A). Altogether, dispersant addition enhanced petroleum hydrocarbon concentrations by a factor of more than 500 (Appendix A). In order to obtain similar starting concentrations for WAF and CE-WAF treatments, the use of dilutions 1:1 for WAF and 1:50 for CE-WAF were found to give a similar order of magnitude (Appendix A).

Distribution of PAH compounds in CE-WAF and WAF treatments also differed. Two-ring PAH compounds (1- and 2- methylnaphthalene and naphthalene) constituted the largest fraction, with 90% in CE-WAF and 98% in WAF (Appendix A). Conversely, three-ring PAH compounds constituted 9% in CE-WAF and 2% in WAF. Four- to six-ring PAH compounds constituted 0.7% in CE-WAF and 0.04% in WAF. Dispersant facilitated higher ring PAHs to be transferred to the water phase.

### 3.2. Assessment of Biodegradation Rates

The highest petroleum hydrocarbon degradation was observed in the coastal GoB seawater WAF treatment (degradation 92%, when concentrations at the end of the experiment are compared with concentrations at the beginning); in the same experiment, CE-WAFs were diluted to 1:1000, leading to concentrations below the limit of detection (<100 µg/L) at the most timepoints. The highest degradation (5%) with dispersant addition was observed in GoF seawater with 1:50 dilution (Figure 2). Compared to experiments with GoF seawater, NS seawater had lower degradation (59%) in WAF treatment, and no degradation was observed with CE-WAF treatment. In the control experiment with GoF seawater, degradation of 40% in WAF and no degradation in CE-WAF were observed (Appendix A).

The measured concentrations of petroleum hydrocarbons increased during the first days of the experiment in the bottles with dispersant (CE-WAF), possibly indicating that the extraction method was not efficient enough with freshly prepared CE-WAF. Only after this increase, a decrease could be observed (Figure 2) in the C_10_-C_40_ concentrations during the experiments. In the control experiment, no reduction of petroleum hydrocarbons was observed (Appendix A).

In WAF treatments, PAH sum concentration was 174 µg/L at the beginning of the experiment (0 h) and decreased to 6.8 µg/L at the end of the experiment (Appendix A). In comparison, in CE-WAF treatments, PAH sum concentration was 159 µg/L at 0 h and 59.57 µg/L at 12 d. PAH degradation was thus much higher in WAF treatment compared to CE-WAF treatment (96% and 60%, respectively).

Part of the observed degradation could be due to evaporation. This was investigated in a control experiment, including abiotic controls. In WAF treatments, lighter compounds (<C16) had decreased when comparing 0 h and 12 d, indicating evaporation. However, when comparing WAF 12 d samples with WAF abiotic 12 d samples, this showed lower spikes in hydrocarbon compounds <C18, indicating possible biodegradation (Figure 3). Based on the concentration of petroleum hydrocarbons obtained, disappearance in the biotic samples (40%) was higher than in abiotic controls (28%), which confirms biodegradation took place in addition to evaporation (Appendix A).

When dispersant alone was added to seawater in the dilution of 1:50, a concentration of 745 µg/L of hydrocarbons C_10_-C_40_ was obtained, and during 12 days of incubation it was reduced by 24% (Appendix A).

Diagnostic ratios (Pristane/Phytane, C17/C18, C17/Pristane, C18/Phytane, Norpristane/Pristane) were calculated for North Sea crude oil, and 12 d CE-WAF and 12 d CE-WAF abiotic control samples (Appendix A). WAF treatment samples of 12 d WAF and 12 d WAF abiotic control were also analysed, but as they did not contain these long-chain biomarker hydrocarbons, diagnostic ratios could not be calculated. Pristane and phytane are branched isoprenoid hydrocarbons that are very resistant to biodegradation and have been commonly used in calculating diagnostic ratios to determine changes in oil composition that are caused by biodegradation [35,36].

All calculated diagnostic ratios were very similar between Northern Sea crude oil and CE-WAF 12 d abiotic control samples, indicating that no biodegradation had occurred in the abiotic control sample. In comparison, C17/Pristane and C18/Phytane ratios in CE-WAF 12 d were lower (2.25 and 4.38) compared to the 12 d CE-WAF abiotic control (1.78 and 3.98) and Northern Sea crude oil (1.88 and 3.98).

### 3.3. Abundance of Hydrocarbon Degradation Related Genes Measured by qPCR

Coastal GoF seawater showed higher abundances of *alkB*, PAH-RHDα and 16S rRNA genes compared to coastal NS and open sea GoB seawater at the beginning of the experiment, reflecting the conditions in situ. During the experiments, the gene copy number of *alkB* genes increased in all samples with oil (Figure 4). The highest PAH-RHDα copy numbers were observed with GoF seawater at the end of the experiments. The PAH-RHDα-GN gene copies were below the detection limit in all samples except for GoF at day 12, where the GN and GP each contributed around 50% to the gene abundance. In the control experiment, higher PAH degradation was observed in WAF treatment compared to CE-WAF treatment at the end of the 12 d experiment (Appendix A). However, the qPCR showed similar PAH gene abundances in both treatments. ANOVA analysis did not show a significant difference between treatments in any of the experiments.

### 3.4. Seawater Bacterial Communities in the Presence of Oil and Dispersant

A principal coordination analysis (PCoA) plot (Figure 5) based on genus level (PCOA plots using OTU level data are presented in Appendix A) data shows that microbial communities from microcosm experiments were different according to the seawater collection location, and this difference was statistically significant according to a PERMANOVA test (R2 = 0.56, *p*-value: 0.001). According to the sample ordination plot, the major difference in bacterial communities was between the Baltic Sea and Norwegian Sea samples.

Ordination of samples on the PCoA plot indicates that the within-group variation was smallest in the case of Norwegian Sea bacterial communities, while the Baltic Sea samples, especially from the Gulf of Finland, varied along the PCoA second axis.

Smaller differences in the microbial community were found between microbial communities in high and low oil GoF experiments (Figure 6, PERMANOVA R2 = 0.25, *p*-value: 0.001) and between different treatments in the NS experiment (Figure 7, R2 = 0.82, *p*-value: 0.001) and GoB experiment (Figure 8, R2 = 0.89, *p*-value: 0.001).

Lefse analysis (Figure 9) indicated that the main differences were that *Flavobacterium* was enriched in GoF seawater, unclassified *Comamonadaceae* in GoF high oil seawater, *Clade Ia* in NS seawater and *RS62_marine_group* and unclassified *Sporichthyaceae* in GoB seawater. Lefse analysis did not reveal significant differences (log LDA score > 2.0) in genus level between different treatments in any of the experiments.

In general, microcosms with GoB and GoF (both high and low oil concentrations) seawater were predominated at genus level by Flavobacterium, unclassified *Comamonadaceae*, *Clade_III_ge*, *Pseudorhodobacter*, *NS11_22_marine group* and *RS62_marine_group* (Figure 10 and Figure 11). A family level bar chart of microbial relative abundances is presented in the Appendix A.

WAF and CE-WAF treatments had apparent effects on the bacterial community structure (Figure 10 and Figure 11), and this was observed especially in the case of GoF seawater experiments at both low and high oil concentrations. In the GoF microcosm with low oil concentration, genus Flavobacterium’s relative abundance increased in the WAF and especially in CE-WAF treatment (average 59% of all communities). Genus *Pseudomonas* increased in the low oil GoF WAF microcosm and CE-WAF treatments (17% and 6%, respectively), but not in the control (1.4%) and in situ samples. In the high oil GoF seawater, the abundance of unclassified *Comamonadaceae* increased (27%) and became the predominant genus in the high oil WAF treatment but decreased slightly in the CE-WAF treatment. Instead, the genus *Pseudomonas* increased more in the CE-WAF treatment (20%) than in WAF (6%).

In microcosms with GoB and NS seawater, there was weaker microbial community response to CE-WAF and WAF treatments compared to GoF experiments. In GoB seawater, *RS62_marine_group* increased during the experiment in WAF and CE-WAF treatments and became the predominant genus, but a similar increase was also observed in control seawater.

The microbial community from NS seawater differed from other microcosms by higher abundances of *Clade Ia*, *NS5_marine_group*, *NS3a_marine_group*, *Polaribacter* and a very low abundance of *Pseudorhodobacter. Clade_Ia* and *NS5_marine_group* decreased in NS seawater in all treatments during the experiment. *NS3a_marine_group* and *Polaribacter*, both belonging to family *Flavobacteriaceae*, increased in the CE-WAF treatment (from 5% to 10% and 4% to 9%, respectively), but remained almost the same or decreased slightly in the WAF and control treatments.

## 4. Discussion

It has been suggested that many laboratory-scale biodegradation experiments on chemically dispersed oil have been conducted using unrealistically high oil concentrations and have not considered the rapid dilution that happens in the sea during real oil spills [7,37]. Based on measurements of real oil spills, Lee et al. (2013) [37] estimated that the average oil concentration below 1 m of the surface of a dispersed oil plume is 100 ppm (=100 mg/L) or less. According to Macnaughton et al. (2003) [38], the oil concentration in dispersed oil plumes is expected to have an average concentration below 10 ppm before biodegradation becomes significant. In wave tank experiments, the immediate oil concentrations have been found to be 5–147 ppm [39]. Li et al. (2009) [40] conducted flow-through experiments of dispersed crude oil with Alaskan North Slope crude oil and Corexit 9500. They observed a decline of dispersed oil from 12 ppm to 2 ppm within 60 min (at a depth of 0.75 m, 10 m downstream from the dispersant application).

Some studies measured oil concentrations from real-life oil spills after the application of dispersants. For example, samples collected during the Deepwater Horizon spill showed only 5% of samples had concentrations higher than 0.25 mg/L, with the highest measured oil concentration of 7270 mg/L [41]. After the Mega Borg spill that occurred in the Gulf of Mexico in 1990, the maximum oil concentration measured under the centre of the slick after dispersant application was 22 mg/L, while the measured concentration on areas without dispersant addition ranged from 1.2 to 3.9 mg/L [42]. During the Sea Empress accident in 1996, several dispersants were sprayed over the oil slick, and the hydrocarbons were measured to be between 3 mg/L and 5 mg/L with dispersant use and 3 mg/L without dispersants at a depth of 5 m. In subsequent measurements, oil concentrations decreased to 0.5–0.6 mg/L after 4 days and to 0.2 mg/L after 12 days [43].

In the present study, we aimed at environmentally relevant petroleum hydrocarbon concentration by using diluted WAFs and CE-WAFs. When studying biodegradation and aquatic ecotoxicity of oil products, it is generally recommended to use these fractions; however, it is seldom that concentrations of both total THP and PAHs are reported in publications, and most often only the 16 EPA PAHs are reported [44]. Undiluted WAFs and CE-WAFs resulted in concentrations of 1000 µg/L and 555,000 µg/L, respectively, which are environmentally unrealistic. In order to obtain relevant levels of petroleum hydrocarbons, the CE-WAF had to be diluted much more than the WAF. Furthermore, the fingerprint of the oil compounds changed more in the WAF in comparison to the original crude oil, with lighter compounds dominating in the WAF. For that reason, it is difficult to design an experiment that investigates solely the effects of the dispersant on oil biodegradation.

Precise measurement of oil concentrations in seawater is associated with many challenges, as due to low solubility, precise measurement of oil concentrations in water is challenging. If oil is present in such a high amount that it is floating on the surface, sampling and measurement of the concentration in water is not reliable. Because oil sticks to the surfaces of sampling bottles, there is a risk of contamination between samples. The presence of dispersant and oil together causes the formation of micelles, and because of this, extraction of oil in micelles may be hampered. Furthermore, there are several methods for extracting and analysing oil. Because oil is a mixture of hundreds of compounds, the analysis result is dependent on what compounds are analysed for. We have used a standardized method (ISO 2000 [20]) for petroleum hydrocarbons in water, which measures the sum of petroleum hydrocarbons with a chain length of C_10_ to C_40_, using GC-FID after extraction with hexane and removal of polar compounds by Florisil. This method has a detection limit of 50 μg L^−1^.

In our study, oil biodegradation was not observed to increase with the addition of dispersant in any of the experiments. On the contrary, the highest oil biodegradation was observed without dispersants in experiments with GoF and GoB experiments. The highest oil concentrations at the beginning of the experiment were ca. 30,000 µL/L (CE-WAF in high oil GoF experiment), and this concentration was most likely toxic for the microbes, as no degradation was observed. In the experiments using diluted WAFs and CE-WAFs, the highest oil concentration was much lower (500–3000 µL/L at the beginning of experiments), and according to the literature cited above, these should not be too high for biodegradation.

Although dispersant did not seem to enhance oil degradation in our experiment when compared to treatments without dispersant, there are some uncertainties when interpreting the results. Different dilution and petroleum hydrocarbon concentrations between WAF and CE-WAF treatments makes it difficult to compare biodegradation rates. Micelles formed by dispersant chemicals might have also affected extraction efficiency, as the chemical analysis method we used is designed for analysing petroleum hydrocarbons from water samples without any interfering compounds. It is also possible that the petroleum hydrocarbon biodegradation is overestimated in our study because part of the petroleum hydrocarbons C_10_-C_40_ may have evaporated or degraded chemically. The disappearance of light compounds was observed in abiotic controls. Differences in chromatogram peaks between biotic and abiotic samples, however, indicated that biodegradation occurred in the biotic samples. Calculated diagnostic ratios C17/Pristane and C18/Phytane for the 12 d CE-WAF were higher compared to the 12 d CE-WAF abiotic control and North Sea crude oil. Typically, a decrease in these ratios during the experiment would suggest biodegradation. However, it seems that not only did easily degradable compounds C16, C17 and C18 decline greatly, but also Phytane and Pristane decreased in the sample. Furthermore, as there were no data from WAF samples to compare, it was difficult to make further conclusions from CE-WAF biodegradation based on the diagnostic ratios.

There was a large variation between the different sites in the biodegradation of petroleum hydrocarbons. The highest degradation was observed in experiments with seawater obtained from the Gulf of Bothnia and the Gulf of Finland followed by the Norwegian Sea.

In this study, no testing was conducted to assess the toxicity of Finasol 51 to bacteria; however, there are a few aquatic and bacterial toxicity studies found in the literature conducted using this substance. In one study [45], Finasol 51 did not generally inhibit marine bacteria growth in plate cultures but clearly inhibited the growth of terrestrial bacteria. In the same study, Finasol 51 was found to be moderately toxic to sea urchin embryos. In another study by Rial et al. (2014) [46] conducted with sea urchins, Finasol 51 was found to significantly contribute to the CE-WAF toxicity. Nikolova et al. (2020) [47] observed that Finasol 52 dispersant led to lower overall microbial diversity but resulted in higher oil biodegradation. We used dispersant Finasol 51 with DOR 1:10, which is a commonly recommended ratio for dispersants and is not expected to be toxic to microbes.

Finasol 51 has also been tested for toxicity in fishes’ early life stages (Johann et al. 2020b). In this study, high energy water accommodated fraction (HE-WAF) treatments containing only Finasol 51 were found to have toxic effects on early-stage fish embryos, but less toxic than CE-WAF treatments containing North Sea crude oil. Finasol 51 was also tested for estrogenicity levels [48], but dispersant alone was not observed to increase estrogenic potential.

Studied *alkB* and PAH degradation gene copy numbers increased towards the end in all experiments, indicating that biodegradation occurred during the experiments. The highest oil degradation gene abundances were observed in microbial communities associated with GoF coastal seawater, and the lowest with GoB seawater. Higher copy numbers found in the experiments with GoF seawater could be explained by higher nutrient content in coastal seawater, although nutrient levels were not measured in this study. Further, this site showed the highest values of 16S rRNA gene, which is an indication that the total number of bacteria was higher at this site. Low nutrient content is known to be a limitation for biodegradation [49], and nutrient stimulation has been observed to increase the abundance of oil-degrading bacteria and oil-degrading genes in cold Arctic seawater [50]. The level of alkane degradation genes (*alkB*) was around 10^3^ times higher than PAH degradation genes (PAH-RHDα), which indicates that there is a higher alkane degradation potential. The number of PAH-RHDα gene copies increased during the experiment in the GoF (low oil) samples at day 12 by a factor of 3. This change is in line with the changes in the community roles, showing that PAH degraders become more abundant later in the oil degradation process. This change is also in accordance with the observed decrease in PAH concentrations in the coastal GoF samples in both WAF and CE-WAF treatments. In addition to bacteria, archaea and fungi have been shown to play a role in the oil degradation process [51,52]. Therefore, studying other oil-biodegradation-related genes could be of importance.

Amplicon sequencing results showed differences in bacterial communities with and without dispersant treatments. This indicates that the bacterial communities change with the addition of oil and dispersant and start to favor petroleum hydrocarbons and dispersant degrading bacteria. Differences were also observed between geographically different seawaters used in the experiments. Lower microbial community response to oil observed with Gulf of Bothnia and Norwegian Sea samples could also be explained by lower nutrient levels limiting the growth of degrading bacteria.

In a diesel microcosm experiment conducted with coastal Baltic Sea seawater [18], *Betaproteobacteria* and *Actinobacteria* or *Alphaproteobacteria* were the dominant bacteria when exposed to diesel, depending on if the sampling site for the seawater was pristine or previously polluted. In another microcosm study by Viggor et al. (2013) with Baltic Sea seawater, *Gammaproteobacteria* (particularly the genus *Pseudomonas*) and *Alphaproteobacteria* were found to be the dominant bacteria when oil is present. In this study, *Betaproteobacteria* genus *Comamonadaceae_unclassified* only dominated in coastal GoF seawater in the presence of high oil concentrations. *Pseudomonas* was found to be dominant in GoF high oil CE-WAF treatment and also increased in GoF low oil WAF treatment.

The main oil-degrading bacteria in Arctic seawater often belong to *Gammaproteobacteria* [53]. Similar results were found in this study, where *Gammaproteobacteria* abundance increased more in the WAF and CE-WAF treatments compared to the control treatment of all microcosms, and the highest abundance was detected in the GoF high oil CE-WAF treatment.

In the literature, *Polaribacter* has been reported to increase in response to oil addition [14]. It has also been observed to respond strongly to dispersant, and there are indications that it is linked with dispersant biodegradation [54]. In this study, the highest abundance of *Polaribacter* was detected in the CE-WAF treatment of the NS microcosm at the end of the experiment, indicating that the increase could be linked with biodegradation of the dispersant.

Previous microcosm studies have indicated that family *Flavobacteriaceae* is linked to oil biodegradation [15]. This taxon was also observed to increase after the beginning of the oil spill, indicating that members of this family consume oil-biodegradation metabolites [55]. *Flavobacteriaceae* has also been reported to enrich in the presence of dispersant Corexit [54,56]. These findings were similar, as although *Flavobacteriaceae* was present in all microcosms and treatments, it was most predominant (60%) in the CE-WAF treatment of the GoF microcosm with low oil concentration, indicating a response to the dispersant when the oil concentration was not too high.

## 5. Conclusions

This study was conducted to evaluate the effects of dispersant on crude oil biodegradation in cold temperatures and on microbial communities. The addition of oil and/or dispersants affected the microbial communities especially in experiments with coastal Gulf of Finland seawater favoring oil-degrading bacteria. Lower response to oil was observed in microbial communities with Gulf of Bothnia and Norwegian Sea seawater, which could be due to possible lower nutrient content found in these seawaters. The dispersant did not seem to enhance oil degradation; however, it is difficult to interpret the results due to uncertainties with chemical analysis and variation in oil concentrations used in the experiments.

## Figures and Tables

**Figure 1 microorganisms-11-00882-f001:**
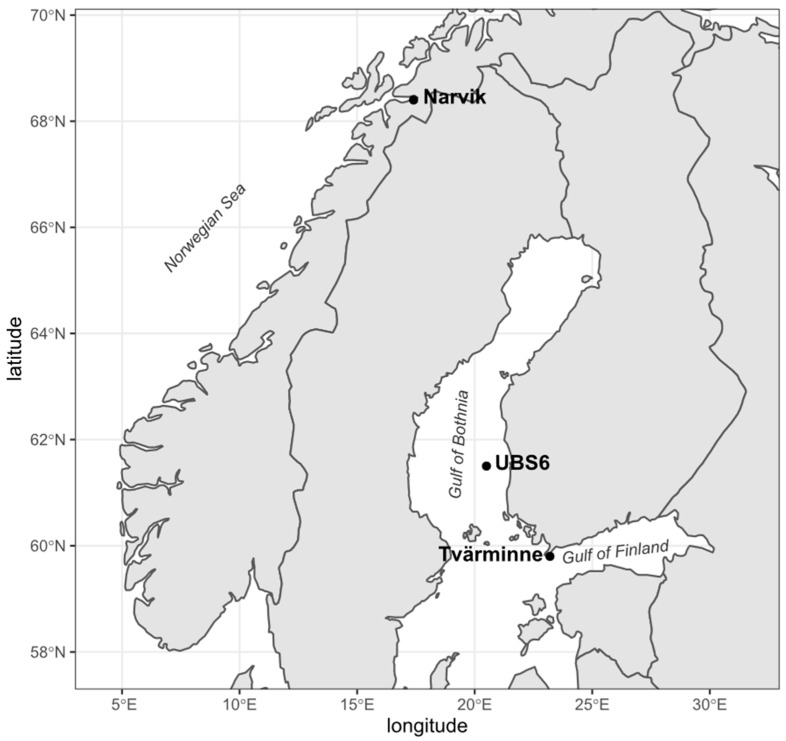
Map of sampling sites. Sampling locations in Tvärminne Zoological Station, UBS6 and Narvik are indicated with black circles.

**Figure 2 microorganisms-11-00882-f002:**
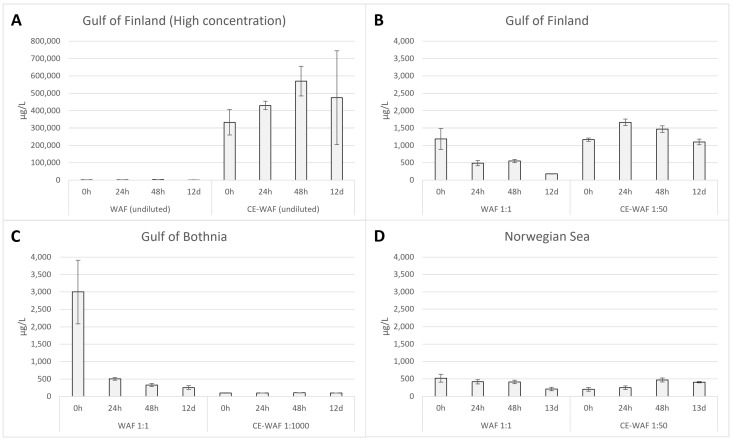
Petroleum hydrocarbon (C_10_-C_40_) concentrations in microcosm experiments with different seawaters (**A**–**D**). Different dilutions used with WAF (without dispersant) and CE-WAF (with dispersant) are presented below the graph; also note that figures have different scales. Shown are mean values (n = 3), and error bars indicate standard deviation. Control samples consisting only of seawater analysed at the beginning of each experiment were always below the limit of detection (<100 µg/L).

**Figure 3 microorganisms-11-00882-f003:**
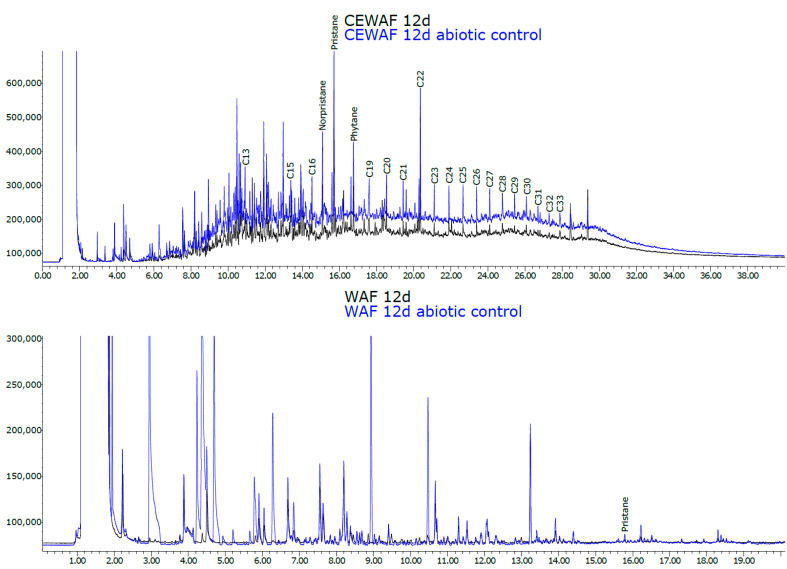
Comparison of gas chromatograms from the control experiment for WAF (without dispersant, dilution 1:1) and CE-WAF (with dispersant, dilution 1:50) at the end of the 12 d experiment. Black color indicates normal 12 d WAF or CE-WAF sample, and blue color indicates abiotic control.

**Figure 4 microorganisms-11-00882-f004:**
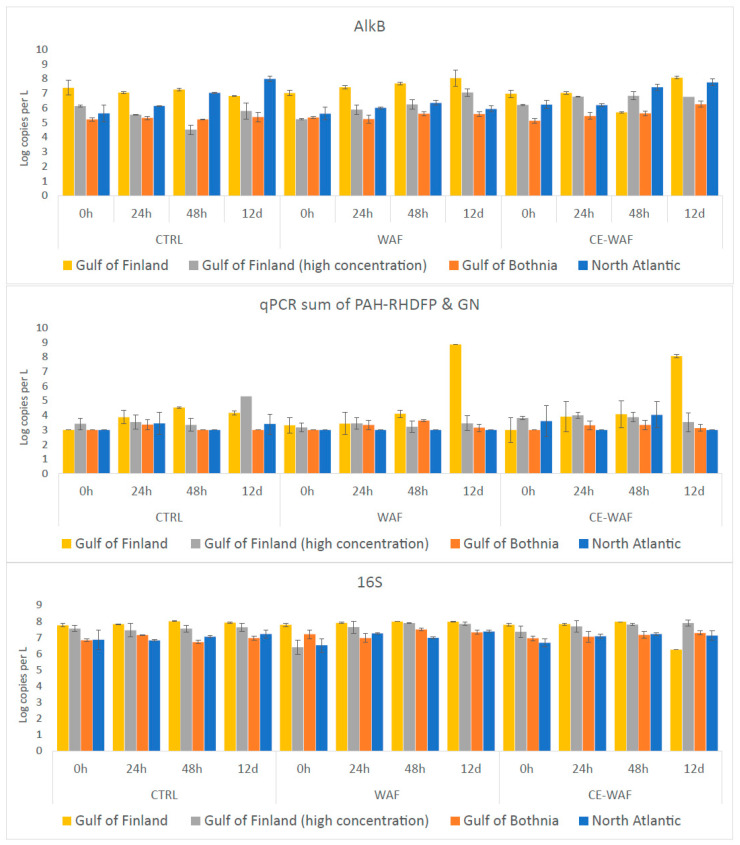
Gene copy numbers of *alkB*, PAH-RHDα (sum of gram-negative and gram-positive gene copies) and 16S rRNA genes for different experiments. Shown are mean values (n = 3), and error bars indicate standard deviation.

**Figure 5 microorganisms-11-00882-f005:**
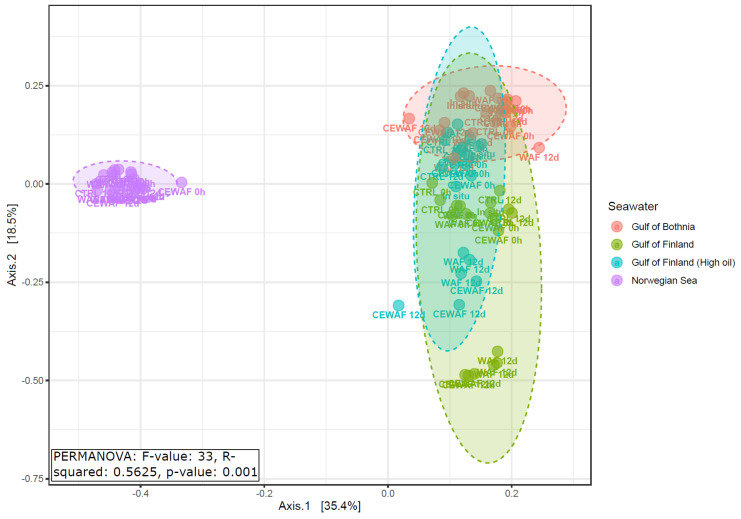
Principal coordinates analysis (PCoA) of microbial communities comparing different microcosm experiments. PCoA was based on a Bray–Curtis distance matrix using genera data.

**Figure 6 microorganisms-11-00882-f006:**
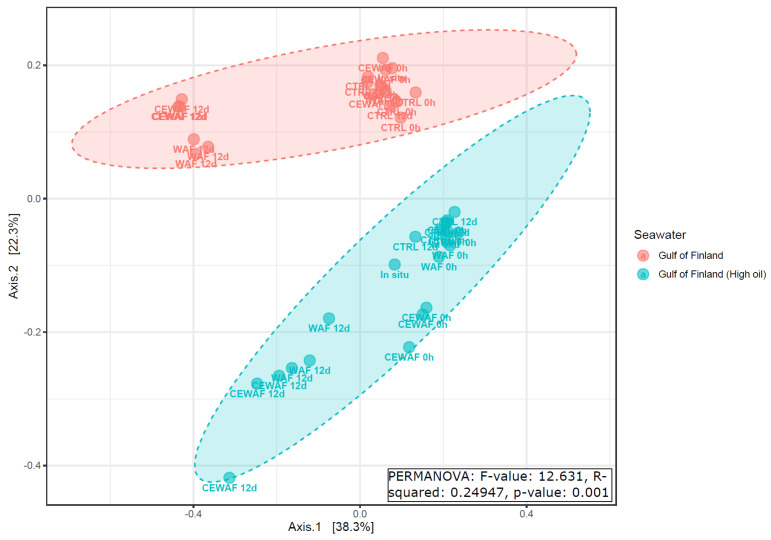
PCoA of microbial communities comparing high and low oil level Gulf of Finland experiments. PCoA was based on a Bray–Curtis distance matrix using genera data.

**Figure 7 microorganisms-11-00882-f007:**
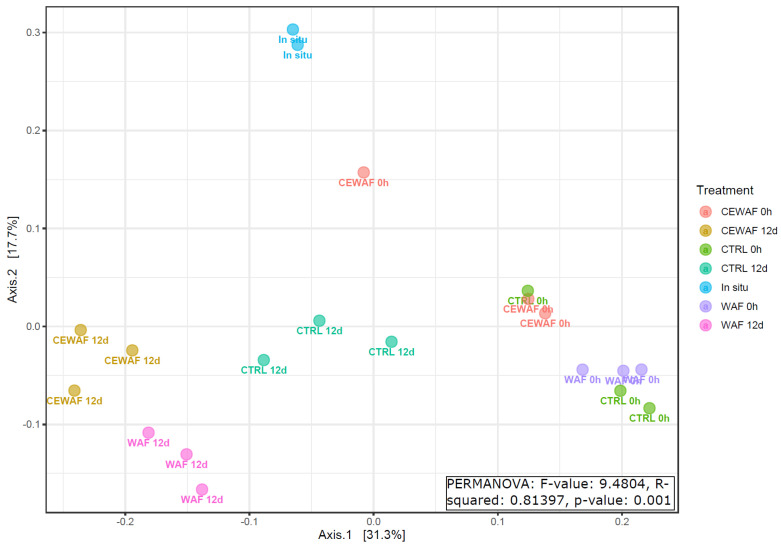
PCoA of microbial communities in the Norwegian Sea experiment. PCoA was based on a Bray–Curtis distance matrix using genera data.

**Figure 8 microorganisms-11-00882-f008:**
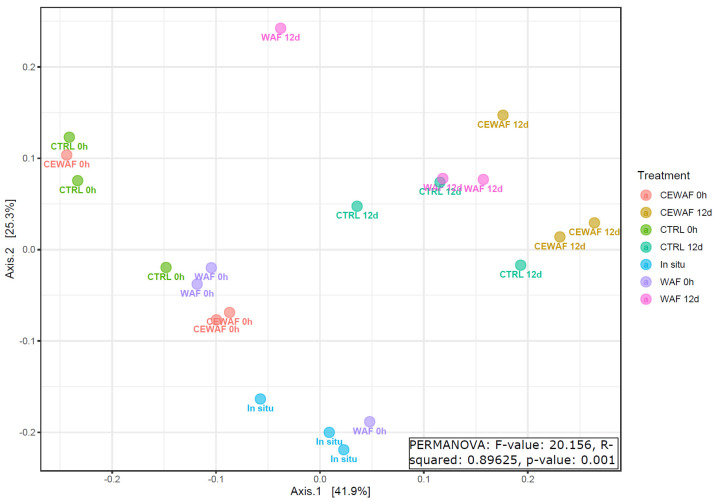
PCoA of microbial communities at the genus level in the Gulf of Bothnia experiment. PCoA was based on a Bray–Curtis distance matrix using genera data.

**Figure 9 microorganisms-11-00882-f009:**
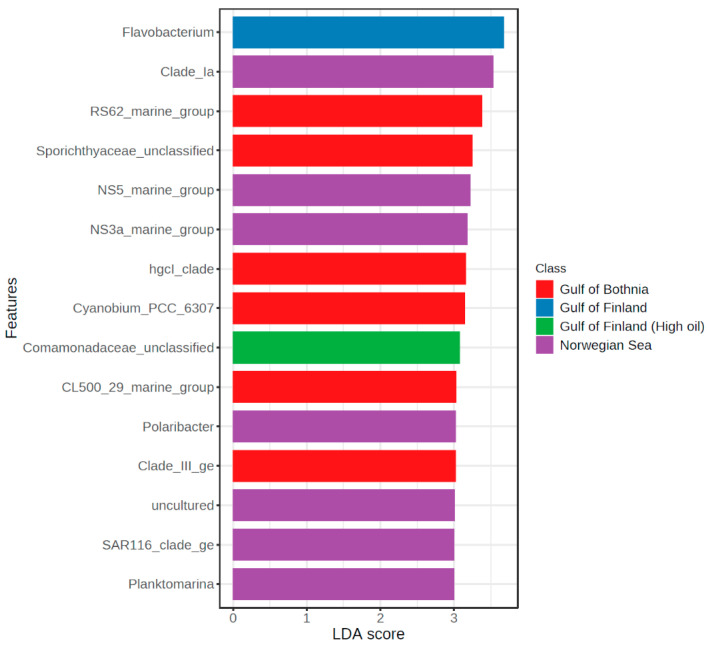
Linear discriminant analysis effect size (LEfSe) analysis of microbial communities comparing seawater from different experiments.

**Figure 10 microorganisms-11-00882-f010:**
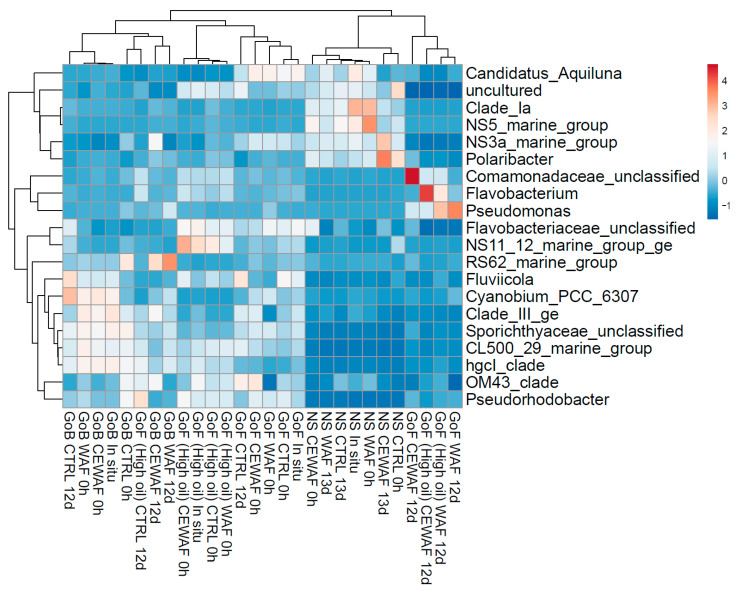
Heatmap of the top 20 most abundant bacterial genera in different experiments and treatments (average of three replicates). NS indicates the Norwegian Sea, GoB indicates the Gulf of Bothnia and GoF indicates the Gulf of Finland.

**Figure 11 microorganisms-11-00882-f011:**
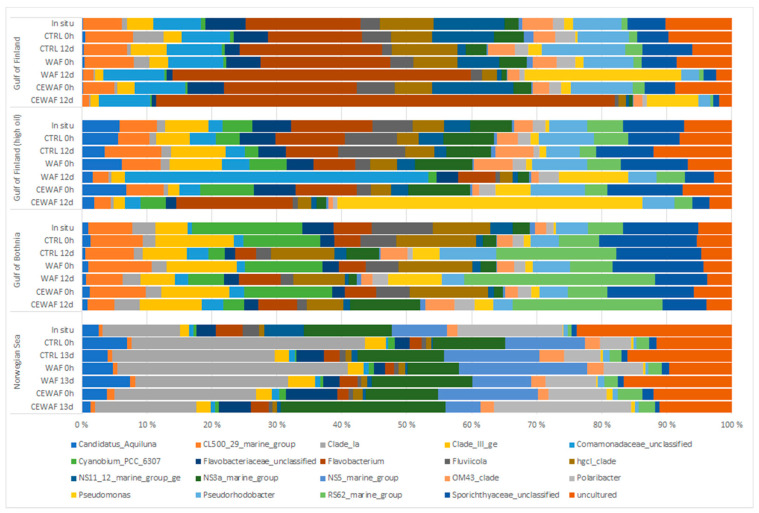
The relative abundance of the top 20 most abundant bacterial taxa at the genus level. Each bar in the figure is the average abundance calculated from three replicates for different experiments (Gulf of Bothnia, Norwegian Sea and Gulf of Finland with high and low oil concentration seawater).

**Table 1 microorganisms-11-00882-t001:** Coordinates of the water sampling sites.

Sampling Location (Code)	Sea Area	Coordinates	Background Information	Date
Tvärminne (GoF)	Gulf of Finland, Baltic Sea	59.8420° N, 23.2018° E	Coastal area, platform (High oil concentration experiment) Coastal area, sampling through ice (Low oil concentration experiment) Coastal area, sampling through ice (Control experiment)	15 May 2017 12 March 2018 7 January 2019
UBS6 (GoB)	Gulf of Bothnia, Baltic Sea	61.5344° N, 20.5433° E	Open sea area, annual monitoring cruise of RV *Aranda*	8 June 2017
Narvik (NS)	Norwegian Sea, North Atlantic	68.4421° N, 17.3892° E	Coastal area	11 June 2018

**Table 2 microorganisms-11-00882-t002:** Overview of samples and dilutions used in different microcosm experiments.

Seawater	Treatments and Dilutions		
	CE-WAF	WAF	Control
GoF high	Undiluted	Undiluted	Only seawater
GoB	1:1000	1:1	Only seawater
GoF low	1:50	1:1	Only seawater
NS	1:50	1:1	Only seawater

**Table 3 microorganisms-11-00882-t003:** Primers used in the qPCR analysis.

Gene	Primer	Primer Sequence	Reference
16S rDNA	Eub338(f)	5′-ACT CCT ACG GGA GGC AGC AG-′3	[25]
	Eub518®	5′-ATT ACC GCG GCT GCT GG-′3	[26]
Gram-positive PAH-RHDɑ	PAH-RHDɑ GN F	5′-GAG ATG CAT ACC ACG TKG GTT GGA-3′	[24]
	PAH-RHDɑ GN R	5′-AGC TGT TGT TCG GGA AGA YWG TGC MGT T-3′	
Gram-negative PAH-RHDɑ	PAH-RHDɑ GP F	5′-CGG CGC CGA CAA YTT YGT NGG-3′	
	PAH-RHDɑ GP R	5′-GGG GAA CAC GGT GCC RTG DAT RAA-3′	
*AlkB*	*AlkB* F	5′-AACTACMTCGARCAYTACGG-3′	[23]
	*AlkB* R	5′-TGAMGATGTGGTYRCTGTTCC-3′	

## Data Availability

Not applicable.

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
