# Peer review of "Effects of Dispersant on the Petroleum Hydrocarbon Biodegradation and Microbial Communities in Seawater from the Baltic Sea and Norwegian Sea"

_microorganisms, 2023, doi:10.3390/microorganisms11040882_

Round 1

Reviewer 1 Report

Review of manuscript 2222159

The manuscript presents results of crude oil biodegradation and changes in bacterial communities with oil contamination in microcosm experiments. The experiments took place in two locations in the Baltic Sea and one in the Norwegian Sea, and lasted for 12 days. The authors attempt to compare oil biodegradation between the sites however this may not be possible due to different starting concentrations of oil between locations. The quantification of oil when dispersant is used was also problematic and no safe data can be extracted from these results either. In all, the biodegradation data should be treated with caution and do not allow strong statements. That means the authors should re-think the title and the overall direction of the paper which is around the effectiveness of dispersant in the studied regions.

The microbial community changes are presented properly in terms of data analysis (although some problems are identified below) however, the results seem random, and we do not see the typical succession patterns or well known hydrocarbon-degraders becoming significantly enriched in the microcosms. Is it because of low oil concentrations in combination with low T? There are several issues that can be discussed.

The main problem of the manuscript is that it is very hard to follow. It feels like a lot of experiments that were conducted without an initial plan were put together without any filter. The narrative becomes even more confusing because one of the experiments is separated from the rest and is named as “control experiment” which was conducted in one of the locations later on and is something different from the controls within each experiment (seawater only) but still includes WAF and CEWAF treatments as in the normal experiments. See lines 254-258 for example: “Highest degradation (number missing but suppose this is a typo) with dispersant addition was observed in GoF seawater with 1:50 dilution.” Then in line 257: “In the control experiment with GoF seawater degradation of 40% in WAF and no degradation in CE-WAF was observed.”. The way this is written, it gives two different results for the same thing and surprisingly they are entirely opposite!

I urge the authors to remove this title (“control experiment”) and simply say that they conducted abiotic controls. The remaining of the “control experiment” (excluding WAF and CEWAF 12d) is simply a study of the behaviour of the oil in WAF and CEWAF i.e., what goes in the water fraction in each case. This is meaningful for undiluted WAF and CEWAF (and possibly diluted just to check that the proportions are the same) and the PAH analysis also makes sense here. But it does not make sense to mention biodegradation of PAHs since this analysis was not performed in the “normal” experiments. I urge the authors to either “forget” the WAF and CEWAF 12d from the “control” experiment or present it together with the rest. Having said that, the starting concentrations in CEWAF in the “control experiment” (10000 ug/L) is much higher than the GoF CEWAF as it stands in Figure 3 (1200 ug/L).

In all, I urge the authors to re-think what exactly serves the narrative of the paper and include only the relevant information. One example is the “control experiment” as mentioned above. Another example: the goal (as mentioned in the discussion) is to emulate environmentally relevant conditions (low conc. of oil). What is the purpose of including a high oil treatment? Also, for treatments to be comparable, you’d need to have comparable oil concentrations. Why include the 1:1000 CEWAF dilution, where hydrocarbon concentrations were below detection? If these are excluded, then the paper might have story and you’d avoid mentioning the dilutions all the time in the text because all would be the same, 1:1 for WAF, 1:50 for CEWAF. Then, as locations you can re-name as: Baltic Sea coastal, Baltic Sea open and Norwegian Sea to keep things simple.    

Detailed comments

Line 14:  replace “brackish” with “low salinity” seawater.

Line 19: “genes involved in hydrocarbons degradation” rather than “hydrocarbon genes”

Line 21: although you have mentioned the locations above, GoB is mentioned here for the first time out of the blue.

Line 79: There should be more background information on the locations such as water temperature or local characteristics. What is different between the coastal and the open sea site in the Baltic? What are the differences between the coastal Baltic Sea and the coastal Norwegian? There would also be differences due to seasonality. While the coastal Baltic experiments (GoF) took place in winter, the Baltic open and Norwegian Sea experiments were conducted in spring. There would be differences, in at least the phytoplankton and nutrient levels which would have an immediate effect on the bacterial community and the degradation rate. Any other information, e.g., pollution levels, in each region are welcome. Did you perform HC quantification of in situ water?

Paragraph 2.4

This paragraph needs to be re-structured according to the general comments above without any mention of an extra “control” experiment. I think I have also spotted a mistake: in line 114, the 1:1000 dilution is mentioned for GoF however, in line 253 the 1:1000 dilution is mentioned for GoB and Figure 3 seems to agree with the latter. Please simplify: a) keep only 1:50 dilutions, b) mention the abiotic controls in line 110 together with the rest of the treatments and remove the rest of the “control experiment”. The authors need to think critically what information they get from each part and if it’s worth adding confusion from numbers that do not offer much to the overall story. Why was the experiment performed again for biotic WAF and CEWAF in the “control experiment”? In Table S1, why is the concentration of hydrocarbons in dispersant only treatment higher than in WAF? What are we looking for in the dispersant only treatment since the constituents of the dispersant are not quantified?

Also, here should be information on how the sampling was performed. It is important to know whether you sub-sampled (and how much) from the microcosms to determine the HC concentrations or you took the whole volume of the 1L bottle. If there are different 1L bottles for chemical analysis and for biological, this should be mentioned. I presume from the inconsistency of the results that sub-sampling was performed. If this is the case, then there is a methodological problem with the HC quantification that cannot be solved at this stage.

A Table in the main text with all treatments and controls would help greatly.

Line 129: First time undiluted WAF and CEWAF from control experiment are mentioned. So far the reader knows about undiluted WAF and CEWAF from GoF abbreviated as GoF high. Again, as far as I understand, these samples were done to check what goes in the water fraction, which is correct and described in 3.1. You can mention that you analysed chemically the WAF and CEWAF before it goes into the treatments in 2.3 and never mention it again as a control or treatment.

Line 146: I was confused here too. So, PAHs were not analysed in the microcosms as for total HC (line 130)? Only in the control experiment? The authors again need to think what the goal is, i.e., PAHs should be analysed in the undiluted WAF and CEWAF and that gives us information of the behaviour of the oil in each treatment. What is the purpose of analysing PAHs in the abiotic controls on 12d, for example, since this measurement was not performed in the microcosms?

Line 192: please mention the region of the 16S gene that this primer pair amplifies.

Line 209: Sequences instead of contigs

Line 219: “differences between treatments”

Line 228: “not all fractions of crude oil were transferred entirely into the…”

Lines 230-231: It’d be good to indicate the regions corresponding to n-alkanes etc on the chromatograms. Figure 2 should be in the Supplementary.

Lines 238-243: Are there any proportional differences between undiluted WAF and 1:1 dilution and undiluted CEWAF and 1:50 dilution? In Table S2, the presence of measurements for day 12 only adds confusion. This Table is useful to see the starting concentrations in each case, not biodegradation.

Line 244: Maybe the title should just be “Assessment of Biodegradation rates”.

Figure 3. Excluding graph A, all the rest should have the same y-axis limits.

Lines 250-258: There is an interesting pattern here which should be thought of. Regardless of the starting concentrations in WAF all your treatments end up with more or less the same amounts of oil after 24h, 48h and 12 d. If you fix the y-axis this would be visible. So then, could this seemingly “higher” degradation in GoB be attributed to the fact that is starts from a higher oil concentration? As for CEWAF, I am not sure how a degradation percentage can be extracted from these measurements. I don’t see  how these measurements can be trusted? Why trust the 0h and 12h and not the inbetween?

Line 263: So, if no biodegradation was observed in CEWAF in the control experiment, which is actually GoF but one year later, then how can the result of highest degradation in CEWAF in GoF be explained? (line 254). I don’t think you can draw any conclusions of CEWAF biodegradation in any area. Also, you have abiotic controls and there seems to be loses from abiotic WAF within 12 d. Are these taken into account and subtracted?

Lines 274-281: the evaluation of abiotic losses should precede those of biodegradation rates and be taken into account (subtracted).

Line 282: Why are there petroleum hydrocarbons in the dispersant only treatment? What does it mean for CEWAF treatments?

Line 287: please use the same terminology throughout, abiotic or sterile.

Line 296: these are actually higher in the 12d CEWAF compared to abiotic CEWAF and crude oil. Shouldn’t it be lower to show biodegradation?

Figure 4.  Same as the other chromatogram. Please indicate on the graph what is said in the text for the non-expert. Where do I find the lighter <16C compounds? Similarly this figure should be in the supplementary.

Line 305-306: I do not see this increase. There is no difference with the control and this is mentioned later with the ANOVA result.

Paragraph 3.4.

Please explain the ordination results altogether at the beginning of the paragraph. Norwegian Sea indeed looks different from the rest. This strengthens the case to re-mane the locations (Baltic coastal, Baltic open, Norwegian). I will not comment on B because my opinion is that the High oil treatment should go. Then B would look similar to C and D. It is not clear what was tested in the PERMANOVA for C and D. A significant result is given but we don’t know the factor that is significant. You should perform PERMANOVA with the full model including the explanatory variables treatment and time and the interaction between them and perform stepwise simplification of the model. You’d need to exclude the in situ samples for that.

I don’t understand how the Lefse analysis was performed. This analysis shows differences between two datasets. In your case this could be the 0h vs. 12h in WAF or CEWAF in each location. Or if WAF and CEWAF are not very different based on PCoA, then these can be pulled together and testing can be done for the time factor in each location. I am not sure what was compared exactly based on lines 335-338 and figure 7.

Figure S1. This should be in the main text in my opinion.

Lines 360-364: This is another example of confusing language; please remove lines 363-364 or rephrase and certainly keep all info about Pseudomonas together.

Lines 370-373: The NS microbial community has been mentioned in lines 342-344. Please keep all info about each location together.

Lines 365-369: The first sentence is what’s important here. Indeed, the response of the microbial community in GoB and NS was almost invisible. The authors need to discuss why that is further down in the discussion. As for the results section, what can be said if there was no response? Some slight increases here and decreases there may be completely random.

Overall, there doesn’t seem to be a consistent pattern with well-known hydrocarbon-degraders becoming enriched in the communities as would be expected. The dominant taxa in each treatment/location combination seem random and the authors need to re-think which samples to use and which to lose to make a story out of this dataset. I have mentioned previously the high oil treatment and the CEWAF in GoB that has undetectable oil concentrations. What does it mean that some controls (e.g., in Norwegian Sea samples) seem to cluster with WAF and CEWAF? Were the bottle effects during incubations stronger than the effect of oil after all?

Discussion. Please keep the units for oil concentrations the same, preferably ppm that is more widely used.

Line 410-412: please rephrase, the sentence is repetitive.

Line 413-414: there is a risk that not all oil is recovered if sub-sampling is practiced, rather than contamination between samples.

Line 423: This statement needs to be re-phrased otherwise this would be one of the few studies that shows halting of degradation with dispersant. The quantification of HCs in CEWAF was problematic, I wouldn’t draw any conclusions based on these results. 

Lines 453: to my opinion any mention of toxicity testing of the dispersant on marine animals is irrelevant to the topic.

Lines 473-474: are there any literature data on nutrients though to support this claim?

Some final suggestions. It would be good to look at the communities at family level too. Maybe some genera are not classified and are thus excluded from analysis? For this reason, I also tend to use the OTU tables for ordination analysis rather than genus level.

Author Response

Comment answers are in the attached file.

Reviewer 2 Report

The study is well designed and I like the concept of filling a niche in an otherwise well-studied area of hydrocarbon biodegradation. I fully agree with the Authors that the Baltic Sea region requires better understanding.
Overall, I have a positive impression regarding the manuscript. The experimental set-up is well planned, the methods were appropriate and reliable, the obtained dataset is diverse and sound in terms of statistics. The Authors included a proper comparison of their results with the reports of other researchers and discussed their findings in a sufficient manner. I have only found some minor aspects (mainly editorial issues) which should be corrected. The comments have been listed below:

Abstract
In its current state, the abstract lacks numeric data (e.g. what was the range of biodegradation efficiencies observed during the study, what was the highest relative abundance for the dominant genus, etc.). Most Readers focus on the Abstract section for crucial data, therefore I strongly advise the Authors to slightly modify this section and include the most important results.

Materials and methods
It would be beneficial to provide the Readers with basic data regarding the contents of the dispersant. While Finasol OSR 51® is very popular, a short note that it includes e.g. a specific type of surfactants will be of high importance for the Readers as this matter has a direct impact on the obtained results.

Results
Figures 2 and 4 – I think that an indication of which compounds are associated with the respective (major) peaks observed in the chromatograms would make the figures more appealing for the Readers. E.g. symbols can be placed above the peaks and a short explanation can be added in the captions.
Figure 6 – it is recommended to either try to fit all four graphs in a single page (which may not be possible as this would deteriorate the clarity) or break this figure into 4 separate graphs with their own captions. It is unacceptable to scroll for two pages just to read the caption.

Conclusions
The manuscript includes a lot of diverse data, which is obviously very good. In order to summarize all the respective elements of the study, I would recommend to include a short Conclusions section with single, bullet-point comments regarding the most important findings.

Minor comments:
The English language in the manuscript is clear and understandable, however there are some minor issues (e.g. the absence of “a” and “the” articles). An example is given below:
Lines 282-284
When dispersant alone was added to seawater in the dilution of 1:50, a concentration of 745 µg/L of hydrocarbons C10-C40 was obtained and during 12 days of incubation it was reduced 24 % (Table S1).
The sentence should be corrected to “(…) it was reduced by 24% (Table S1)”.
It is recommended to thoroughly read the manuscript and correct the contents, preferably with the help of a native speaker in order to further improve its quality.

Author Response

(The authors gave the same response as above.)

Reviewer 3 Report

Oil spills have severe effects on the ambient area around the release site. These events could be caused by mishaps that happen during storage or transit, as well as during oil and gas exploration or production-related activities.  Microorganisms' ability to break down oil can be used to mitigate some of the immediate concerns related to oil spills with little negative environmental impact. The use of microcosms for the biodegradation of crude oil in challenging environments, such as high-salinity sea water, subfreezing temperatures, and the presence of a surfactant, is rare, though. Hence, there is a lot of interest in this manuscript. The problem is made clear in the introduction, which also outlines the goal of the studt. In-depth and logical descriptions of the materials and methods are provided. The findings are well-discussed.

There are only minor comments:

- Figure 5: Please separate ctrl, waf and cewaf at regular intervals so that the columns in the barcharts don't overlap and the image is clear.

Author Response

(The authors gave the same response as above.)

Round 2

Reviewer 1 Report

The authors have made changes to improve the clarity of the manuscript and this is appreciated. I liked the conclusions section and the fact that it is honest. As stated, no safe conclusions can be drawn on biodegradation based on this dataset for various methodological and analytical reasons. This honesty should be present from the beginning in the abstract, and throughout. 
